# Influenza Vaccine Uptake in Italy—The 2022–2023 Seasonal Influenza Vaccination Campaign in Italy: An Update from the OBVIOUS Project

**DOI:** 10.3390/vaccines12030297

**Published:** 2024-03-12

**Authors:** Angelo Capodici, Aurelia Salussolia, Giusy La Fauci, Zeno Di Valerio, Marco Montalti, Anna Odone, Claudio Costantino, Heidi J. Larson, Julie Leask, Jacopo Lenzi, Lamberto Manzoli, Davide Gori

**Affiliations:** 1Unit of Hygiene, Public Health and Medical Statistics, Department of Biomedical and Neuromotor Sciences, University of Bologna, 40126 Bologna, Italyaurelia.salussolia@studio.unibo.it (A.S.); zeno.divalerio@studio.unibo.it (Z.D.V.); marco.montalti7@studio.unibo.it (M.M.); jacopo.lenzi2@unibo.it (J.L.); lamberto.manzoli2@unibo.it (L.M.); davide.gori4@unibo.it (D.G.); 2Interdisciplinary Research Center for Health Science, Sant’Anna School of Advanced Studies, 56127 Pisa, Italy; 3Department of Public Health, Experimental and Forensic Medicine, University of Pavia, 27100 Pavia, Italy; anna.odone@unipv.it; 4Department of Health Promotion Sciences, Maternal and Infant Care, Internal Medicine and Excellence Specialties “G. D’Alessandro”, University of Palermo, 90133 Palermo, Italy; claudio.costantino01@unipa.it; 5Infectious Disease Epidemiology, London School of Hygiene & Tropical Medicine Institute of Health Metrics, London WC1E 7HT, UK; heidi.larson@lshtm.ac.uk; 6Institute for Health Metrics and Evaluation, University of Washington, Seattle, WA 98195, USA; 7School of Public Health, Faculty of Medicine and Health, University of Sydney, Sydney 2006, NSW, Australia; julie.leask@sydney.edu.au

**Keywords:** vaccine, uptake, influenza, flu, health policy

## Abstract

Influenza is a significant public health concern, with Italy being profoundly impacted annually. Despite extensive vaccination campaigns and cooperative initiatives between the Public Health Departments of Local Healthcare Authorities and family physicians, low vaccine uptake rates persist. This study builds upon the OBVIOUS project, providing an updated picture of influenza vaccine uptake in Italy through a representative sample. A cross-sectional computer-assisted web interviewing (CAWI) survey of 10,001 Italian citizens was conducted between 31 March and 5 June 2023. Our findings underscore the negative impact of a lack of awareness that a person is in a priority group for influenza vaccination (−26.1 percentage points in vaccine uptake) and the profound influence of social circles on vaccination decisions (−5 percentage points when unfavorable). Medical professionals played a pivotal role, with recommendations from family doctors significantly promoting vaccine uptake (+20.2 percentage points). Age, chronic conditions, and socio-demographic factors also influenced vaccination behaviors. For children, parental negative perceptions regarding the flu (−10.4 percentage points) and vaccine safety (−23.4 percentage points) were crucial determinants. The present study emphasizes the need for a comprehensive approach addressing awareness, societal beliefs, and tailored medical advice to enhance vaccination rates and protect public health in Italy.

## 1. Introduction

Influenza remains a public health challenge that requires continuous vigilance. It consistently impacts a significant portion of Italy’s population, leading to severe health consequences, particularly among specific vulnerable groups [1,2]. Across the broader European Union landscape, this disease results in a considerable number of preventable deaths annually [3].

Last year, in the first survey of the OBVIOUS project (Observatory on Vaccine Hesitancy in Italy—Online UniBo Surveys), we underscored the pivotal role of vaccination as the primary defense against influenza disease, as well as other viruses [4,5]. With the World Health Organization (WHO) and the new Italian National Vaccine Prevention Plan advocating for high coverage targets, Italy has undertaken commendable steps to provide free vaccinations for those at an elevated risk of influenza-induced complications [6,7].

These efforts are exemplified by the collaborative initiatives between the Public Health Departments of Local Healthcare Authorities (LHAs) and family physicians, ensuring optimal vaccination distribution, particularly within Italy’s decentralized healthcare framework.

Nevertheless, despite these structured approaches, vaccine uptake remains a multifaceted issue, affected by individual beliefs, societal pressures, and logistical challenges [8,9]. Furthermore, scientific evidence suggests that different factors such as age, gender, healthcare utilization or accessibility, education, income, socioeconomic status, and types of chronic diseases have a relevant impact on vaccine uptake, especially among older people [10].

Our prior research highlighted the enduring challenge of vaccine uptake and hesitancy, which the WHO has spotlighted as a significant barrier [11]. Previous studies showcased sub-optimal vaccination rates across different segments of the Italian population, indicating an evident gap between the targeted and actual vaccination coverage [12,13].

Furthermore, recent studies across EU countries reveal critical insights into the uptake and policies surrounding influenza vaccination. Key findings demonstrate a suboptimal vaccination rate among adults, especially those with pre-existing health conditions [14]. Despite a general trend towards supporting the co-administration of COVID-19 and influenza vaccines, data show significant variability in vaccine coverage across different regions, with Eastern Europe lagging behind Western and Northern Europe and North America [15]. The studies also highlight the diverse approaches to vaccine recommendations across countries like Germany, Spain, the UK, and Italy, ranging from strict evidence-based criteria to more flexible real-world evidence considerations [16]. This underscores the challenge of harmonizing vaccine recommendations across the EU. The data suggest an urgent need for better coordination and the sharing of best practices to enhance vaccine uptake and protect public health.

As we move forward, it is imperative to re-evaluate and understand the evolving landscape of influenza vaccine hesitancy and uptake in Italy. This article aims to provide an updated perspective on the social and behavioral factors influencing influenza vaccination. Through this endeavor, we aspire to further equip policymakers and health professionals with nuanced, evidence-based findings to inform future strategies.

## 2. Materials and Methods

The study used a cross-sectional design employing a computer-assisted web interviewing (CAWI) questionnaire. Data collection spanned from 31 March to 5 June 2023, with the assistance of Dynata (https://www.dynata.com/ accessed on 4 March 2024). A sample of 10,001 Italian citizens aged 18 years and older participated in the survey, using a stratified sampling method. This technique involved proportionate allocation based on first-level NUTS (Nomenclature of Territorial Units for Statistics) region of residence (Northwest, Northeast, Central, South, and Islands), gender, and age group, categorized as 18–24, 25–34, 35–44, 45–54, 55–64, and 65 years and above.

The survey, designed for a 10 min completion, featured six sections. Prior to full implementation, a soft launch was conducted to assess its effectiveness and gather feedback, which informed subsequent revisions and improvements to the questionnaire. The first section investigated participants’ background information, including the following: gender; age; region and province of residence; educational attainment; occupation; living arrangements; ability to pay for essential items; whether they have children and, if so, the gender and age of the youngest child and shared responsibility for their vaccinations; and clinical data, such as pregnancy, daily living difficulties due to physical or mental impairments, body mass index (BMI), chronic respiratory diseases, cardiovascular diseases, and diabetes. Other questions in the section addressed topics such as the place where most vaccinations were received, the preferred place, time slot, and day of the week for receiving vaccines, friends’ and family’s views on vaccination, and the primary source of information for recommended vaccines.

Sections #2 to #6 were dedicated to collecting data on vaccination against influenza (the focus of this work), pneumococcus, herpes zoster, meningococcus, and tetanus. To align with the recommendations of the Italian Ministry of Health, the subsample selected to answer the influenza section met any of the following criteria: aged 60 years or above; pregnant women in October/November 2022; individuals with underlying clinical conditions such as chronic respiratory diseases, chronic cardiovascular diseases, diabetes, or a BMI of 30 kg/m^2^ or above; individuals exposed to occupational risks, encompassing healthcare workers, teachers, and law-enforcement members; or parents or guardians of children aged between 6 months and 6 years. Questions were tailored according to a predefined set of priority criteria: respondents meeting at least one of the criteria applicable to the adult population were directed to respond on their own behalf, while those not meeting these criteria were prompted to provide details regarding the vaccination status of their youngest child.

In the influenza section, questions related to the following topics were asked: being vaccinated against influenza (three response options: “yes”, “no”, “not sure”); worry about getting sick with seasonal influenza (“not worried”, “a little worried”, “quite worried”, “very worried”); vaccine safety perception (“very safe”, “quite safe”, “quite unsafe”, “very unsafe”); awareness of being in the target population for the vaccination program (“yes”, “no”, “don’t know”); advice they would give to friends or relatives invited for vaccination (“get it”, “don’t get it”, “don’t know”); who invited them for vaccination (“LHA”, “family doctor/pediatrician”, “occupational doctor”, “gynecologist/obstetrician”, “other specialists”, “no one”); ease in accessing vaccination for those who received it (“very easy”, “quite easy”, “quite difficult”, “very difficult”); and knowing how to receive the vaccine for those who did not (“yes”, “no”). Those who answered that they had not been vaccinated against influenza were asked about their current intention to receive the vaccine (“yes” or “no”). Parents or guardians were specifically instructed to answer all the questions on behalf of their offspring. The questionnaire translated into English can be found in the Appendix A.

Data management for the survey conducted by Dynata followed the General Data Protection Regulation (GDPR) of the European Union (EU), ensuring the protection and privacy of personal data of individuals within the EU. Additionally, the survey adhered to all relevant requirements outlined by Italian regulations, ensuring that data collection, storage, and processing were conducted in accordance with applicable laws and guidelines in Italy. These measures were implemented to safeguard the privacy and confidentiality of participants’ information throughout the survey process.

### Statistical Analysis

Post-stratification by gender, age group, and area of residence confirmed that non-response to the survey within certain strata of Italy’s population was negligible, exerting no substantial impact on the study’s overall sample estimates of 10,001 individuals (results not presented). Consequently, adjusting sampling weights for the targeted subsample of respondents for influenza vaccination (*n* = 5788) was deemed unnecessary.

All data were summarized using counts and percentages or mean values ± standard deviation (SD), and were visually represented with bar charts, square charts, and thematic maps featuring superimposed pie charts. Estimates were stratified by gender and NUTS region, as well as according to the specific characteristics defining the sample, such as age 60 years or older, children, pregnant women, individuals with respiratory diseases, cardiovascular diseases, diabetes, and/or a BMI of 30 kg/m^2^ or above, medical doctors (MDs), other healthcare workers (nurses, pharmacists, optometrists, etc.), teachers, and law-enforcement members (LEMs).

Multivariable multinomial logistic regression analysis was conducted to explore the determinants of a three-category nominal outcome: “I did get the vaccine”, “I did not get the vaccine, but I would”, and “I did not get the vaccine, and I would not”. Employing a single multinomial logistic model instead of a series of binary logistic models ensured the efficiency and consistency of estimators across study outcomes. Consistent with the vaccination framework outlined by the BeSD Expert Working Group [8], the covariates considered as potential drivers of vaccine uptake, delay, and refusal were as follows: attitudes and beliefs regarding influenza infection and vaccination (perceived worry, safety concerns, and advice from acquaintances on vaccination); social processes (gender and views on vaccination); and practical issues (awareness of higher vaccination priority and invitation source for vaccination). Additional sociodemographic determinants considered were age group, NUTS region, level of urbanization, educational attainment, and clinical/professional factors conferring higher vaccination priority.

The concept of “motivation”, a component of the BeSD framework, could not be measured because intentions were combined with behavior (vaccination) in our outcome variable, and therefore, it could not be tested as a mediator in the association of beliefs and social processes with vaccine uptake.

The effect of covariates was assessed by examining the marginal effect of changing their values on the average predicted probability of observing each outcome. The marginal effect was computed as a discrete difference in probabilities (Δ), with 95% confidence intervals (CIs) obtained using the delta method. Covariate categories occurring in less than 5% of the sample were combined with adjacent lower or upper classes to improve the stability and efficiency of the regression estimates. The Small–Hsiao test for assessing the independence of irrelevant alternatives (IIA) did not indicate the need for alternative model specifications, such as nested logistic models wherein binary logit coefficients fail to converge in probability towards the same values as the multinomial logit coefficients.

All analyses were carried out with Stata 18 [17], and were separately conducted for individuals responding on their own behalf and those responding on behalf of their offspring. No issues of multicollinearity were found in the regression analysis; the variance inflation factor was below 5, and the condition index was below 10 for each covariate.

## 3. Results

### 3.1. Sample’s Sociodemographic Characteristics 

Of the 5788 respondents identified as targets for influenza vaccination, 134 (2.3%) were excluded due to their inability to recall their own (*n* = 109) or their children’s (*n* = 25) vaccination statuses. As a result, the remaining 5217 adults who provided data about their own experiences and preferences, and 437 adults who provided data about their children, were included in the analysis.

The sociodemographic characteristics of the respondents who provided details about their own seasonal influenza vaccine uptake are detailed in Table 1. The breakdown showed that there were slightly more females (50.9%) than males (49.0%), with a small representation of non-binary individuals (0.1%). The mean age of the respondents was 54.8 ± 15.9 years, with 38.8% being 65 years or older. The majority resided in either cities (43.5%) or towns and suburbs (43.1%). When considering educational background, over half had a high school diploma (54.6%) and 30.3% had an academic or higher degree. The occupations varied, with the largest segments being retirees (36.3%) and those in occupations other than the ones investigated by our questionnaire (31.7%). The sociodemographic characteristics of those who reported their children’s influenza vaccination statuses can be found in Appendix A.

### 3.2. Seasonal Influenza Vaccination Status

To delineate the geographical variations in seasonal influenza vaccination uptake from October to December 2022, data detailing vaccination acquisition, delay, and refusal were stratified by NUTS statistical region. As represented in Table 2, out of the respondents who answered on their own behalf, 45.4% received the vaccine, 8.2% did not but were willing to, and 46.3% did not and had no intention of doing so. Northeastern Italy recorded the highest vaccine uptake at 52.2%, whereas Insular Italy exhibited the lowest figure at 39.3%. Central Italy displayed the highest resistance to the influenza vaccine at 48.9%, while the Northeast showed the least resistance at 39.1%.

Figure 1 provides insights for each high-risk target group based on age, clinical status, and profession, while Figure 2 provides the breakdown for vaccine uptake in the population as a whole. Individuals aged 60 years and above reported a vaccination rate of 53.0%, while 6.1% were unvaccinated but willing to do so, and 40.9% abstained with no intention of getting vaccinated. When looking at parents’ responses on behalf of their children, the lowest uptake was observed at 27.9%. In professional categories, vaccine uptake ranged from a low of 29.4% among LEMs to a high of 64.1% for medical doctors. Lastly, among clinical at-risk groups, we registered the lowest uptake among those with a BMI of 30 kg/m^2^ or more (36.6%), and the highest among those diagnosed with diabetes (61.2%).

Gender-specific stratifications of each target group can be referred to in Appendix A.

### 3.3. Perceptions about Influenza and Its Vaccines

The data shown in Figure 3 highlight concerns about contracting seasonal influenza within various high-risk groups. Individuals aged 60 and above showed low levels of concern, with 73.7% reporting being either “not worried” or “a little worried”. A similar trend was observed among children, as reported by their parents, with 66.6% expressing low apprehension (ranging from “not worried” to “a little worried”), as well as among pregnant women (63.3%, ranging from “not worried” to “a little worried”). The most worried categories were people suffering from diabetes, respiratory diseases, or cardiovascular diseases, with 39.1%, 37.6%, and 37.3%, respectively, reporting being “quite worried” to “very worried”.

Turning to the perception of the safety of seasonal influenza vaccines in Appendix A, the overall perception of influenza vaccine safety was positive, with 79.0% of respondents reporting their belief that the vaccine was safe, ranging from “very safe” to “quite safe”. Appendix A provides insights into the perception of vaccine safety among at-risk groups.

### 3.4. Knowledge of Being in a Priority Group, Information Attitudes, and Preferences

Figure 4 shows the awareness of at-risk groups concerning their prioritization for the seasonal influenza vaccination. Seniors (60 years and above) were the most informed, with 78.1% acknowledging their higher priority. The community with diabetes displayed a heightened sense of their vulnerability, with 75.1% aware of their priority status. Similarly, people with cardiovascular diseases and respiratory diseases showed high awareness at 74.8% and 69.1%, respectively. Conversely, the understanding diminished for children, with only 51.0% of their parents being aware. Pregnant women lacked awareness in 52.2% of cases.

Table 3 provides insight into the entities or healthcare professionals who invited different high-risk groups for seasonal influenza vaccination during the last quarter of 2022. For seniors aged 60 and above, the most frequent invitations came from family doctors (58.5%). In contrast, 29.3% claimed that no one had reached out to them. For children, pediatricians invited vaccination in 41.0% of cases, yet 42.3% of children’s parents reported no invitation at all. Pregnant women had more varied invitation sources. While family doctors were predominant at 33.5%, occupational doctors and gynecologists/obstetricians were also notable sources at 16.5% and 9.2%, respectively. However, 27.5% reported no invitation.

People with diabetes relied heavily on family doctors at 57.0%. Yet, 13.7% of this group reported receiving no invitation. Similarly, people with cardiovascular diseases and respiratory diseases reported invitation rates of 55.8% and 51.0% from family doctors, respectively. The no-invitation rates for these groups stood at 20.8% and 20.9%, respectively.

Table 4 provides insights into the vaccination habits and preferences of respondents, segmented by NUTS statistical region. When discussing locations for vaccination habits, vaccine hubs emerged as the predominant choice, with 66.3% of Italians, especially in the Northwest (70.1%), opting for this venue. Regarding the most favored places for vaccination, again, vaccine hubs were preferred by 40.6% of respondents, a figure particularly pronounced in the Northwest (44.1%). However, family doctors were reported as the second favorite place to receive the vaccine nationwide (28.7%), especially in the Islands (33.5%). Timing-wise, the period between 9:00 and 12:00 a.m. was the most favored slot, with 45.0% of preferences. Early morning hours, specifically 6:00–9:00 a.m., were less popular. When it came to days, Monday was the preferred day for vaccination (29.4%), especially in Central Italy (32.0%), while Sunday witnessed the least inclination with a 4.2% preference nationally.

### 3.5. Multivariable Regression Analysis

As presented in Table 5, an absence of awareness about priority for seasonal influenza vaccination exhibited a significant association with increased likelihood of both vaccine refusal and delayed acceptance. Other factors emerged as independent and significant predictors of vaccine refusal; these encompassed age between 35 and 64 years, a diminished concern about contracting the flu, perceptions of seasonal influenza vaccination as unsafe, opposition from acquaintances towards vaccination, and a willingness to dissuade friends and family from getting vaccinated against influenza. On the contrary, significant predictors of vaccine uptake were being 65 years old or older, having chronic cardiovascular diseases or diabetes, and receiving vaccination invitations from the LHA, family doctor, or other medical specialists. Additionally, there were other factors with significant but less substantial effect sizes, including the NUTS region of residence, the degree of urbanization, and educational attainment.

When analyzing vaccine behavior among children (Appendix A), we identified significant predictors of vaccine refusal, which included a lack of concern about children contracting the flu, a perception of seasonal influenza vaccination as unsafe, and a willingness to discourage friends and family from getting vaccinated against influenza. On the other hand, we observed that receiving vaccination invitations from the LHA, a pediatrician, or other medical specialists significantly increased vaccine uptake. Additionally, residing in Northeastern Italy and being unaware that children have higher vaccination priority were significantly associated with lower vaccine uptake.

## 4. Discussion

Our updated analysis provides an understanding of the determinants and barriers influencing the uptake of seasonal influenza vaccination, expanding upon the findings of the initial OBVIOUS project [4]. The self-reported vaccination uptake of 45.4%, while comparable to last year’s 45.7%, suggests a stagnation in vaccination rates. This plateauing could be attributed to persistently being unaware of being prioritized for the vaccine, saturation points in outreach efforts, and persistent hesitancy. The increase in refusal rate (from 33.2% to 46.3%) is a concerning trend. This may indicate a potential rise in skepticism about the vaccine or the health system as a whole [18,19], or misinformation about the influenza vaccine [19], but it may also relate to post-pandemic disengagement from vaccination more generally. In addition, patterns observed in geographical variations regarding vaccine uptake showed to be consistent between last year’s iteration and this year’s, pointing out deeply rooted regional differences in healthcare engagement. This persistence suggests that localized public health strategies might be more effective than a one-size-fits-all approach, and, further, approaches based on established theoretical models might also enhance the effectiveness of such strategies [20].

One salient observation is the negative impact of a lack of awareness about prioritization for seasonal influenza vaccination, which increases the likelihood of vaccine refusal (+25.2 percentage points). This underscores the importance of information dissemination, specifically tailored to communicate the significance of the vaccine to specific demographic groups, especially those aged 35 to 64 years; furthermore, these data indicate the need to have HCWs recommend influenza vaccination more frequently and to have a reminder system in place [21,22]. This demographic indeed showed strong tendencies towards vaccine refusal, just like the previous year, primarily if they were not concerned about contracting the flu or if they perceived the vaccine as unsafe.

It is worth mentioning that the influence of social circles—including close family and friends—appears to be a double-edged sword. While loved ones opposing vaccination significantly contributed to refusal rates (+5.4 percentage points), on the brighter side, receiving vaccination invitations from trusted medical sources, such as family doctors, emerged as robust predictors for vaccine uptake (+20.2 percentage points). This underscores the vital role that medical professionals play in shaping vaccination decisions. Their recommendations carry weight, emphasizing the need for medical professionals to be well informed and proactive in advising patients about the importance and safety of vaccinations. This is especially important considering how much alternative mediums of information are linked to vaccination refusal [23].

Our analysis further indicates that those aged 65 and above, as well as individuals with chronic cardiovascular diseases or diabetes, showed a higher propensity for vaccine acceptance. This is understandable, given that these groups are at a higher risk of complications from influenza, emphasizing the importance of focused campaigns for these at-risk groups.

The role of socio-demographic factors in determining vaccination behavior cannot be understated. Our findings suggest that NUTS region of residence, degree of urbanization, and educational attainment, while significant, had a relatively lesser influence on the decision-making process. It is imperative to delve deeper into understanding these nuances to tailor strategies that resonate with various demographics [24].

Transitioning to vaccination behavior among children, several factors stand out. The lack of concern about children contracting the flu and perceptions of vaccine safety are significant hurdles, as substantiated by Chan et al. [25]. Various studies have also shown how often parents do not consider pediatric influenza serious enough to vaccinate their young [26,27]; our results fall in line with this already-reported trend. Furthermore, our results indicate that even pregnant women express low apprehension about getting sick with influenza, which might be the reason hindering their uptake. Given the influence of parents and guardians in decisions regarding vaccinations, targeted campaigns addressing these specific concerns, and correct information regarding influenza consequences, are paramount.

Moreover, our data accentuate the importance of proactive outreach. A direct invitation from a pediatrician or other medical specialists substantially boosted the likelihood of vaccine uptake among children. However, a geographical divide is evident, with children in Northeastern Italy exhibiting lower vaccination rates. Understanding regional disparities, perhaps in terms of healthcare infrastructure, public health campaigns, or socio-cultural differences, can provide insights into refining national strategies.

In conclusion, our analysis underscores the importance of localized, targeted strategies to address the stagnation in influenza vaccination rates. Increasing refusal rates highlight the need for effective communication to counter skepticism and misinformation. Leveraging medical professionals’ influence and tailoring outreach to demographic-specific concerns are essential for overcoming vaccine hesitancy and improving public health outcomes.

### Strengths and Limitations

This study presents comprehensive data gathered from a large national sample of individuals in Italy who were recommended and offered seasonal influenza vaccination. The sample was categorized into three groups: those who received the vaccine, those who did not receive it but would consider it, and those who did not receive it and would not consider it. This three-fold classification allowed us to better understand the various levels of vaccine hesitancy and explore the underlying factors contributing to reluctance in vaccination.

Additionally, we introduced new relevant covariates to the previous OBVIOUS survey [4,5], such as advice from friends and family and the source of vaccination invitation. However, this study is subject to certain limitations. Firstly, due to its cross-sectional nature, it can only establish statistical associations between variables and does not allow for causal inferences. Secondly, despite efforts to ensure the sample’s representativeness of Italy’s demographics, the online paid survey format may have attracted participants with technological skills seeking additional income. This could potentially result in an over-representation of both lower socioeconomic classes and higher levels of educational attainment. Thirdly, the reliance on self-reported data introduces the possibility of reporting bias. Additionally, household income, religion, and other sensitive social characteristics were not explored because these factors could have influenced the size, power, and representativeness of the sample. Furthermore, immunocompromised individuals, despite their prioritization by the Italian Ministry of Health, were not investigated, and there were missing data regarding the gestational age of pregnant women. Moreover, for the sake of brevity, selection bias may have been introduced in the survey design, particularly when analyzing data related to children. Questions related to transportation, which could be a barrier to vaccination for some individuals, were also omitted. Lastly, lifestyle factors such as smoking, low physical activity levels, inadequate diet, and alcohol consumption appear to be negatively associated with vaccine uptake [10]. Future research is warranted to explore specific lifestyle factors that could influence vaccine uptake in the Italian population, which could inform further strategies to increase influenza vaccination uptake.

## Figures and Tables

**Figure 1 vaccines-12-00297-f001:**
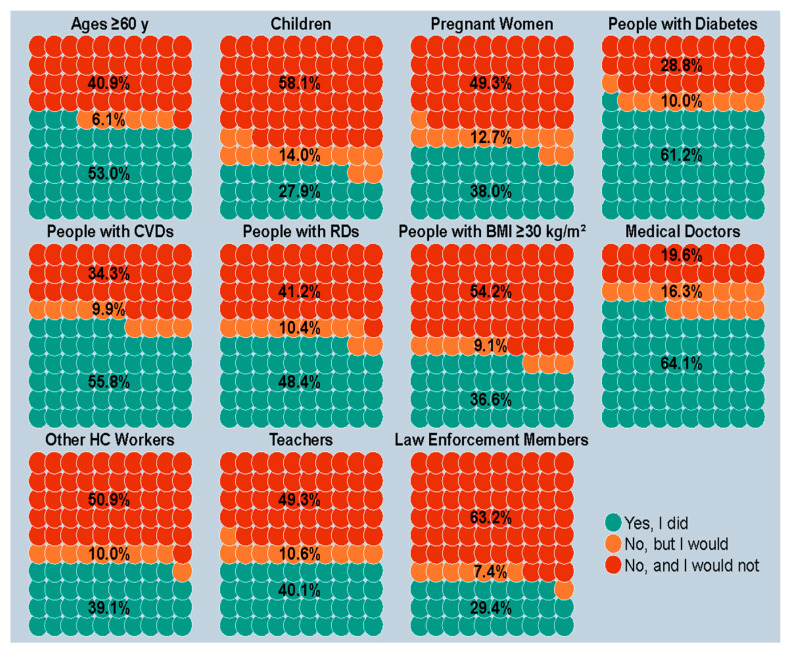
Influenza vaccine uptake between October and December 2022, stratified by high-risk target groups based on age, clinical condition, and profession; if the answer was no, the participants were queried regarding their willingness to receive it. *Notes*: Information regarding children was supplied by their parents. Among the 108 pregnant women who got vaccinated, 35 (32.4%) reported a mean gestational age of 12.5 ± 11.8 weeks at time of vaccination, while 73 (67.6%) were unaware of pregnancy status. *Abbreviations*: CVD, cardiovascular disease; RD, respiratory disease; BMI, body mass index; HC, healthcare.

**Figure 2 vaccines-12-00297-f002:**
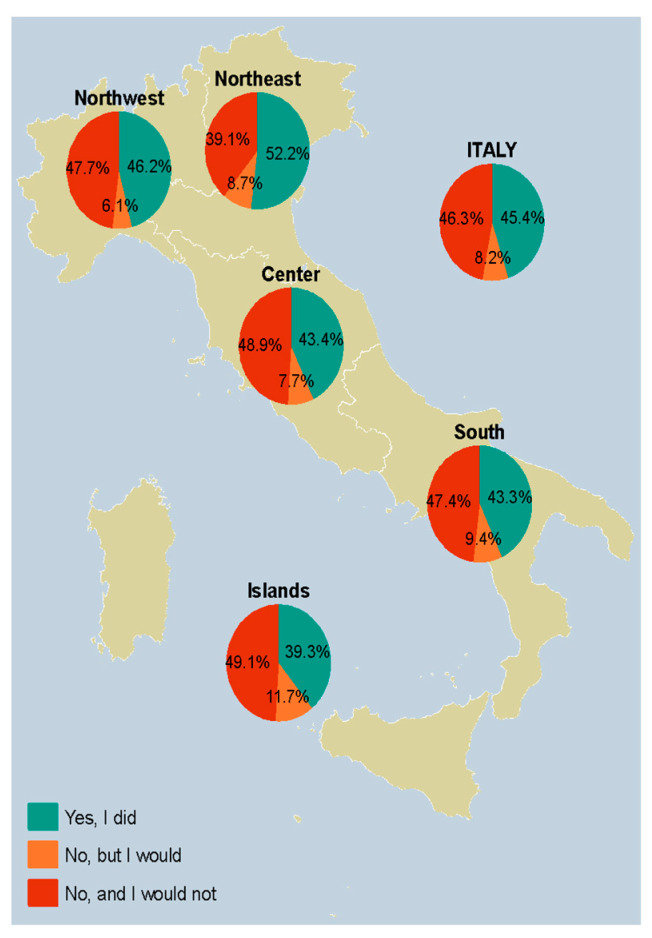
Influenza vaccine uptake between October and December 2022 among participants responding on their own behalf (*n* = 5217) (overall and stratified by NUTS region); if the answer was no, the participants were queried regarding their willingness to receive it. *Notes*: The Northwest comprises Piedmont, Aosta Valley, Lombardy, and Liguria; the Northeast comprises Trentino-South Tyrol, Veneto, Friuli-Venezia Giulia, and Emilia-Romagna; Central Italy comprises Tuscany, Umbria, Marche, and Lazio; the South comprises Abruzzo, Molise, Campania, Apulia, Basilicata, and Calabria; the Islands comprise Sicily and Sardinia. *Abbreviations*: NUTS, Nomenclature of Territorial Units for Statistics.

**Figure 3 vaccines-12-00297-f003:**
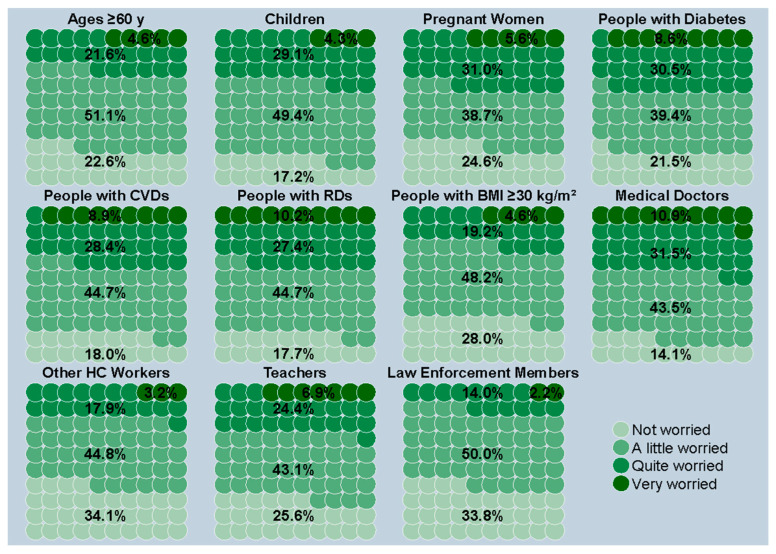
Worry about getting sick with influenza, stratified by high-risk target groups based on age, clinical status, and profession. *Notes*: Information regarding children was supplied by their parents. *Abbreviations*: CVD, cardiovascular disease; RD, respiratory disease; BMI, body mass index; HC, healthcare.

**Figure 4 vaccines-12-00297-f004:**
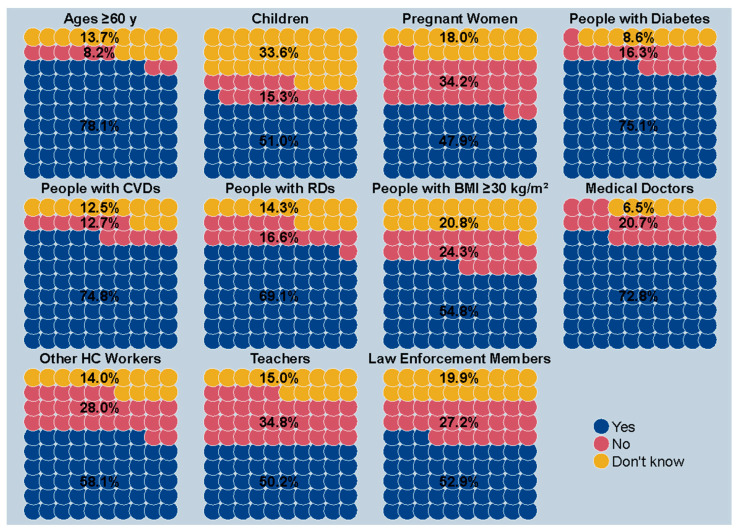
Awareness of being assigned higher priority for influenza vaccination, stratified by high-risk target groups based on age, clinical condition, and profession. *Notes*: Information regarding children was supplied by their parents. *Abbreviations*: CVD, cardiovascular disease; RD, respiratory disease; BMI, body mass index; HC, healthcare.

**Table 1 vaccines-12-00297-t001:** Sociodemographic characteristics of the participants who provided information about their own influenza vaccine uptake (overall and stratified by NUTS region).

Characteristic	Italy	Northwestern Italy	Northeastern Italy	Central Italy	Southern Italy	Insular Italy
	(*n* = 5217)	(*n* = 1517)	(*n* = 998)	(*n* = 1011)	(*n* = 1151)	(*n* = 540)
Gender						
Male	2557 (49.0%)	690 (45.5%)	572 (57.3%)	465 (46.0%)	588 (51.1%)	242 (44.8%)
Female	2653 (50.9%)	825 (54.4%)	425 (42.6%)	545 (53.9%)	560 (48.7%)	298 (55.2%)
Non-binary	7 (0.1%)	2 (0.1%)	1 (0.1%)	1 (0.1%)	3 (0.3%)	0 (0.0%)
Age, y						
Mean ± SD	54.8 ± 15.9	58.3 ± 14.7	52.0 ± 18.4	55.3 ± 15.1	52.9 ± 15.5	53.4 ± 15.1
Age group, y						
18–24	270 (5.2%)	52 (3.4%)	145 (14.5%)	24 (2.4%)	36 (3.1%)	13 (2.4%)
25–34	487 (9.3%)	88 (5.8%)	92 (9.2%)	100 (9.9%)	146 (12.7%)	61 (11.3%)
35–44	682 (13.1%)	147 (9.7%)	111 (11.1%)	145 (14.3%)	198 (17.2%)	81 (15.0%)
45–54	603 (11.6%)	142 (9.4%)	92 (9.2%)	113 (11.2%)	161 (14.0%)	95 (17.6%)
55–64	1149 (22.0%)	367 (24.2%)	176 (17.6%)	262 (25.9%)	227 (19.7%)	117 (21.7%)
≥65	2026 (38.8%)	721 (47.5%)	382 (38.3%)	367 (36.3%)	383 (33.3%)	173 (32.0%)
Place of residence degree of urbanization						
City (densely populated area)	2270 (43.5%)	641 (42.3%)	439 (44.0%)	427 (42.2%)	570 (49.5%)	193 (35.7%)
Town or suburb (intermediate-density area)	2250 (43.1%)	694 (45.7%)	418 (41.9%)	426 (42.1%)	414 (36.0%)	298 (55.2%)
Rural area (thinly populated area)	697 (13.4%)	182 (12.0%)	141 (14.1%)	158 (15.6%)	167 (14.5%)	49 (9.1%)
Educational attainment						
Less than high school diploma	784 (15.0%)	282 (18.6%)	144 (14.4%)	124 (12.3%)	137 (11.9%)	97 (18.0%)
High school diploma	2848 (54.6%)	872 (57.5%)	502 (50.3%)	559 (55.3%)	605 (52.6%)	310 (57.4%)
Academic degree	957 (18.3%)	227 (15.0%)	145 (14.5%)	224 (22.2%)	270 (23.5%)	91 (16.9%)
Post-graduate/doctorate degree	628 (12.0%)	136 (9.0%)	207 (20.7%)	104 (10.3%)	139 (12.1%)	42 (7.8%)
Occupation						
Teacher	434 (8.3%)	111 (7.3%)	60 (6.0%)	83 (8.2%)	124 (10.8%)	56 (10.4%)
Healthcare worker (excl. medical doctor)	279 (5.3%)	75 (4.9%)	38 (3.8%)	60 (5.9%)	65 (5.6%)	41 (7.6%)
Law-enforcement member	136 (2.6%)	21 (1.4%)	31 (3.1%)	25 (2.5%)	38 (3.3%)	21 (3.9%)
Student	110 (2.1%)	30 (2.0%)	14 (1.4%)	17 (1.7%)	33 (2.9%)	16 (3.0%)
Medical doctor	92 (1.8%)	22 (1.5%)	15 (1.5%)	17 (1.7%)	28 (2.4%)	10 (1.9%)
Other occupation	1654 (31.7%)	424 (27.9%)	391 (39.2%)	354 (35.0%)	345 (30.0%)	140 (25.9%)
Unemployed	617 (11.8%)	154 (10.2%)	58 (5.8%)	116 (11.5%)	182 (15.8%)	107 (19.8%)
Retired	1895 (36.3%)	680 (44.8%)	391 (39.2%)	339 (33.5%)	336 (29.2%)	149 (27.6%)
Household composition						
Alone	849 (16.3%)	302 (19.9%)	184 (18.4%)	169 (16.7%)	124 (10.8%)	70 (13.0%)
Couple	3383 (64.8%)	984 (64.9%)	677 (67.8%)	623 (61.6%)	774 (67.2%)	325 (60.2%)
Family of origin	689 (13.2%)	159 (10.5%)	94 (9.4%)	137 (13.6%)	195 (16.9%)	104 (19.3%)
Other	296 (5.7%)	72 (4.7%)	43 (4.3%)	82 (8.1%)	58 (5.0%)	41 (7.6%)
Able to pay for things needed in life						
With great difficulty	666 (12.8%)	157 (10.3%)	90 (9.0%)	139 (13.7%)	181 (15.7%)	99 (18.3%)
With some difficulty	2200 (42.2%)	628 (41.4%)	381 (38.2%)	445 (44.0%)	478 (41.5%)	268 (49.6%)
Quite easily	1913 (36.7%)	592 (39.0%)	405 (40.6%)	362 (35.8%)	406 (35.3%)	148 (27.4%)
Easily	438 (8.4%)	140 (9.2%)	122 (12.2%)	65 (6.4%)	86 (7.5%)	25 (4.6%)

*Notes*: The Northwest comprises Piedmont, Aosta Valley, Lombardy, and Liguria; the Northeast comprises Trentino-South Tyrol, Veneto, Friuli-Venezia Giulia, and Emilia-Romagna; Central Italy comprises Tuscany, Umbria, Marche, and Lazio; the South comprises Abruzzo, Molise, Campania, Apulia, Basilicata, and Calabria; the Islands comprise Sicily and Sardinia. *Abbreviations*: NUTS, Nomenclature of Territorial Units for Statistics; SD, standard deviation.

**Table 2 vaccines-12-00297-t002:** Influenza vaccine uptake between October and December 2022 among participants responding on their own behalf (overall and stratified by gender and NUTS region); if the answer was no, the participants were queried regarding their willingness to receive it.

	All	Yes, I Did	No, but I Would	No, and I Would
				Not
Males and females				
Italy	5217	2371 (45.4%)	429 (8.2%)	2417 (46.3%)
Northwestern Italy	1517	701 (46.2%)	93 (6.1%)	723 (47.7%)
Northeastern Italy	998	521 (52.2%)	87 (8.7%)	390 (39.1%)
Central Italy	1011	439 (43.4%)	78 (7.7%)	494 (48.9%)
Southern Italy	1151	498 (43.3%)	108 (9.4%)	545 (47.4%)
Insular Italy	540	212 (39.3%)	63 (11.7%)	265 (49.1%)
Males				
Italy	2557	1251 (48.9%)	223 (8.7%)	1083 (42.4%)
Northwestern Italy	690	348 (50.4%)	46 (6.7%)	296 (42.9%)
Northeastern Italy	572	334 (58.4%)	60 (10.5%)	178 (31.1%)
Central Italy	465	214 (46.0%)	29 (6.2%)	222 (47.7%)
Southern Italy	588	260 (44.2%)	60 (10.2%)	268 (45.6%)
Insular Italy	242	95 (39.3%)	28 (11.6%)	119 (49.2%)
Females				
Italy	2660	1120 (42.1%)	206 (7.7%)	1334 (50.2%)
Northwestern Italy	827	353 (42.7%)	47 (5.7%)	427 (51.6%)
Northeastern Italy	426	187 (43.9%)	27 (6.3%)	212 (49.8%)
Central Italy	546	225 (41.2%)	49 (9.0%)	272 (49.8%)
Southern Italy	563	238 (42.3%)	48 (8.5%)	277 (49.2%)
Insular Italy	298	117 (39.3%)	35 (11.7%)	146 (49.0%)

*Notes*: Females include non-binary persons. The Northwest comprises Piedmont, Aosta Valley, Lombardy, and Liguria; the Northeast comprises Trentino-South Tyrol, Veneto, Friuli-Venezia Giulia, and Emilia-Romagna; Central Italy comprises Tuscany, Umbria, Marche, and Lazio; the South comprises Abruzzo, Molise, Campania, Apulia, Basilicata, and Calabria; the Islands comprise Sicily and Sardinia. *Abbreviations*: NUTS, Nomenclature of Territorial Units for Statistics.

**Table 3 vaccines-12-00297-t003:** Entities extending invitations for influenza vaccination between October and December 2022, stratified by high-risk target groups based on age, clinical condition, and profession.

	LHA	Family Doctor	Occupational Doctor	Gynecologist/ Obstetrician	Other Specialists	No One
Ages ≥ 60 y	103 (3.6%)	1660 (58.5%)	158 (5.6%)	0 (0.0%)	85 (3.0%)	833 (29.3%)
Children	35 (8.0%)	179 (41.0%)	34 (7.8%)	0 (0.0%)	4 (0.9%)	185 (42.3%)
Pregnant women	29 (10.2%)	95 (33.5%)	47 (16.5%)	26 (9.2%)	9 (3.2%)	78 (27.5%)
People with diabetes	116 (12.2%)	542 (57.0%)	132 (13.9%)	7 (0.7%)	24 (2.5%)	130 (13.7%)
People with CVDs	51 (8.0%)	356 (55.8%)	68 (10.7%)	5 (0.8%)	25 (3.9%)	133 (20.8%)
People with RDs	53 (9.9%)	274 (51.0%)	73 (13.6%)	7 (1.3%)	18 (3.4%)	112 (20.9%)
People with BMI ≥ 30 kg/m^2^	64 (5.5%)	578 (49.4%)	112 (9.6%)	4 (0.3%)	35 (3.0%)	378 (32.3%)
Medical doctors	18 (19.6%)	28 (30.4%)	20 (21.7%)	2 (2.2%)	8 (8.7%)	16 (17.4%)
Other HC workers	24 (8.6%)	111 (39.8%)	67 (24.0%)	2 (0.7%)	16 (5.7%)	59 (21.1%)
Teachers	36 (8.3%)	230 (53.0%)	57 (13.1%)	1 (0.2%)	14 (3.2%)	96 (22.1%)
Law-enforcement members	7 (5.1%)	50 (36.8%)	47 (34.6%)	0 (0.0%)	8 (5.9%)	24 (17.6%)

*Notes*: Information regarding children was supplied by their parents. *Abbreviations*: LHA, Local Healthcare Authority; CVD, cardiovascular disease; RD, respiratory disease; BMI, body mass index; HC, healthcare.

**Table 4 vaccines-12-00297-t004:** Behaviors and preferences for receiving influenza vaccines, social influence, and primary sources of recommendation among participants who provided information about their own influenza vaccine uptake (overall and stratified by NUTS region).

Characteristic	Italy	Northwestern Italy	Northeastern Italy	Central Italy	Southern Italy	Insular Italy
	(*n* = 5217)	(*n* = 1517)	(*n* = 998)	(*n* = 1011)	(*n* = 1151)	(*n* = 540)
Place where you prevalently receive vaccines						
Vaccine hub	3459 (66.3%)	1064 (70.1%)	634 (63.5%)	625 (61.8%)	776 (67.4%)	360 (66.7%)
Hospital	713 (13.7%)	218 (14.4%)	148 (14.8%)	152 (15.0%)	126 (10.9%)	69 (12.8%)
Family doctor	711 (13.6%)	147 (9.7%)	128 (12.8%)	173 (17.1%)	176 (15.3%)	87 (16.1%)
Workplace	129 (2.5%)	31 (2.0%)	41 (4.1%)	19 (1.9%)	29 (2.5%)	9 (1.7%)
Pharmacy	144 (2.8%)	44 (2.9%)	30 (3.0%)	33 (3.3%)	28 (2.4%)	9 (1.7%)
Home	61 (1.2%)	13 (0.9%)	17 (1.7%)	9 (0.9%)	16 (1.4%)	6 (1.1%)
Favorite place to receive vaccines						
Vaccine hub	2118 (40.6%)	669 (44.1%)	395 (39.6%)	393 (38.9%)	468 (40.7%)	193 (35.7%)
Family doctor	1496 (28.7%)	331 (21.8%)	298 (29.9%)	334 (33.0%)	352 (30.6%)	181 (33.5%)
Hospital	726 (13.9%)	219 (14.4%)	143 (14.3%)	141 (13.9%)	144 (12.5%)	79 (14.6%)
Pharmacy	442 (8.5%)	190 (12.5%)	61 (6.1%)	76 (7.5%)	83 (7.2%)	32 (5.9%)
Home	288 (5.5%)	68 (4.5%)	55 (5.5%)	49 (4.8%)	75 (6.5%)	41 (7.6%)
Workplace	147 (2.8%)	40 (2.6%)	46 (4.6%)	18 (1.8%)	29 (2.5%)	14 (2.6%)
Favorite time slot to receive vaccines						
6:00–9:00 a.m.	681 (13.1%)	171 (11.3%)	139 (13.9%)	129 (12.8%)	181 (15.7%)	61 (11.3%)
9:00–12:00 a.m.	2346 (45.0%)	684 (45.1%)	402 (40.3%)	462 (45.7%)	536 (46.6%)	262 (48.5%)
12:00–3:00 p.m.	680 (13.0%)	199 (13.1%)	196 (19.6%)	104 (10.3%)	115 (10.0%)	66 (12.2%)
3:00–6:00 p.m.	963 (18.5%)	295 (19.4%)	172 (17.2%)	216 (21.4%)	193 (16.8%)	87 (16.1%)
6:00–9:00 p.m.	547 (10.5%)	168 (11.1%)	89 (8.9%)	100 (9.9%)	126 (10.9%)	64 (11.9%)
Favorite day of the week to receive vaccines						
Monday	1535 (29.4%)	420 (27.7%)	285 (28.6%)	324 (32.0%)	347 (30.1%)	159 (29.4%)
Tuesday	769 (14.7%)	259 (17.1%)	142 (14.2%)	141 (13.9%)	148 (12.9%)	79 (14.6%)
Wednesday	824 (15.8%)	268 (17.7%)	201 (20.1%)	128 (12.7%)	156 (13.6%)	71 (13.1%)
Thursday	496 (9.5%)	143 (9.4%)	128 (12.8%)	77 (7.6%)	104 (9.0%)	44 (8.1%)
Friday	622 (11.9%)	188 (12.4%)	106 (10.6%)	126 (12.5%)	137 (11.9%)	65 (12.0%)
Saturday	754 (14.5%)	181 (11.9%)	105 (10.5%)	164 (16.2%)	210 (18.2%)	94 (17.4%)
Sunday	217 (4.2%)	58 (3.8%)	31 (3.1%)	51 (5.0%)	49 (4.3%)	28 (5.2%)
Friends and family’s views on vaccination						
Very unfavorable	264 (5.1%)	79 (5.2%)	43 (4.3%)	50 (4.9%)	59 (5.1%)	33 (6.1%)
Unfavorable	188 (3.6%)	51 (3.4%)	33 (3.3%)	28 (2.8%)	55 (4.8%)	21 (3.9%)
Quite unfavorable	532 (10.2%)	151 (10.0%)	99 (9.9%)	101 (10.0%)	122 (10.6%)	59 (10.9%)
Quite favorable	1632 (31.3%)	430 (28.3%)	320 (32.1%)	311 (30.8%)	393 (34.1%)	178 (33.0%)
Favorable	1430 (27.4%)	428 (28.2%)	271 (27.2%)	291 (28.8%)	298 (25.9%)	142 (26.3%)
Very favorable	1171 (22.4%)	378 (24.9%)	232 (23.2%)	230 (22.7%)	224 (19.5%)	107 (19.8%)
Primary source of information to know recommended vaccines						
Family doctor	3386 (64.9%)	1014 (66.8%)	617 (61.8%)	702 (69.4%)	719 (62.5%)	334 (61.9%)
TV	542 (10.4%)	136 (9.0%)	105 (10.5%)	88 (8.7%)	158 (13.7%)	55 (10.2%)
Internet	524 (10.0%)	148 (9.8%)	102 (10.2%)	89 (8.8%)	116 (10.1%)	69 (12.8%)
Healthcare workers (excl. family doctors)	350 (6.7%)	115 (7.6%)	54 (5.4%)	70 (6.9%)	66 (5.7%)	45 (8.3%)
Friends/relatives	214 (4.1%)	46 (3.0%)	57 (5.7%)	33 (3.3%)	58 (5.0%)	20 (3.7%)
Journals	201 (3.9%)	58 (3.8%)	63 (6.3%)	29 (2.9%)	34 (3.0%)	17 (3.1%)

*Notes*: The Northwest comprises Piedmont, Aosta Valley, Lombardy, and Liguria; the Northeast comprises Trentino-South Tyrol, Veneto, Friuli-Venezia Giulia, and Emilia-Romagna; Central Italy comprises Tuscany, Umbria, Marche, and Lazio; the South comprises Abruzzo, Molise, Campania, Apulia, Basilicata, and Calabria; the Islands comprise Sicily and Sardinia. *Abbreviations*: NUTS, Nomenclature of Territorial Units for Statistics.

**Table 5 vaccines-12-00297-t005:** Findings from multivariable multinomial logistic regression analysis: determinants of influenza vaccine uptake and hesitancy, characterized as delay vs. refusal, among participants responding on their own behalf (*n* = 5217).

Characteristic	Did Receive the Vaccine			Would Receive the Vaccine			Would Not Receive the Vaccine		
	Predicted	Discrete Difference (Δ)		Predicted	Discrete Difference (Δ)		Predicted	Discrete Difference (Δ)	
	Probability	Estimate	95% CI	Probability	Estimate	95% CI	Probability	Estimate	95% CI
Gender									
Male	45.5%	Ref.		8.8%	Ref.		45.7%	Ref.	
Female ^†^	45.4%	0.0	−2.1, 2.1	7.7%	−1.1	−2.6, 0.4	46.9%	1.2	−0.8, 3.1
Age group, y									
18–34	42.1%	Ref.		14.0%	Ref.		43.9%	Ref.	
35–44	39.8%	−2.3	−6.5, 1.9	8.6%	−5.4 *	−8.6, −2.2	51.6%	7.7 *	4.0, 11.4
45–54	39.4%	−2.6	−7.2, 1.9	10.0%	−4.0 *	−7.5, −0.4	50.5%	6.6 *	2.7, 10.6
55–64	43.4%	1.4	−2.6, 5.3	8.2%	−5.8 *	−8.9, −2.6	48.3%	4.4 *	1.0, 7.9
≥65	51.1%	9.0 *	5.2, 12.9	5.5%	−8.5 *	−11.5, −5.5	43.4%	−0.5	−3.9, 2.9
NUTS statistical region									
Northwestern Italy	46.3%	Ref.		6.9%	Ref.		46.8%	Ref.	
Northeastern Italy	48.8%	2.5	−0.6, 5.5	7.6%	0.7	−1.4, 2.8	43.7%	−3.1 *	−5.9, −0.3
Central Italy	45.3%	−1.1	−4.1, 2.0	8.5%	1.6	−0.6, 3.8	46.2%	−0.5	−3.3, 2.2
Southern Italy	43.7%	−2.6	−5.5, 0.3	8.8%	1.9	−0.2, 3.9	47.5%	0.7	−2.0, 3.4
Insular Italy	41.2%	−5.1 *	−8.8, −1.4	11.3%	4.4 *	1.6, 7.3	47.5%	0.7	−2.7, 4.1
Degree of urbanization									
City	47.2%	Ref.		7.8%	Ref.		45.0%	Ref.	
Town or suburb	44.2%	−2.9 *	−5.1, −0.7	8.4%	0.6	−1.0, 2.2	47.3%	2.3 *	0.3, 4.4
Rural area	43.3%	−3.9 *	−7.2, −0.6	9.2%	1.4	−1.0, 3.8	47.5%	2.5	−0.5, 5.5
Educational attainment									
Post-graduate/doctorate degree	47.9%	Ref.		8.7%	Ref.		43.4%	Ref.	
Academic degree	48.6%	0.6	−3.2, 4.4	6.9%	−1.8	−4.3, 0.8	44.5%	1.1	−2.5, 4.8
High school diploma	44.6%	−3.3	−6.9, 0.2	8.0%	−0.7	−3.1, 1.8	47.4%	4.0 *	0.6, 7.4
Less than high school diploma	42.4%	−5.6 *	−9.9, −1.2	10.7%	2.0	−1.3, 5.3	46.9%	3.5	−0.5, 7.6
Occupation									
Teacher	45.2%	Ref.		9.2%	Ref.		45.6%	Ref.	
Healthcare worker (incl. MD)	48.3%	0.0	−4.0, 3.9	8.9%	1.1	−1.8, 3.9	42.7%	−1.0	−4.7, 2.6
Law-enforcement member	42.6%	3.0	−1.0, 7.1	6.4%	0.8	−2.0, 3.7	51.0%	−3.9 *	−7.6, −0.2
Other	45.3%	−2.7	−9.3, 3.9	8.1%	−1.7	−5.7, 2.2	46.6%	4.4	−1.7, 10.5
Pneumopathy									
Yes	46.3%	1.0	−2.4, 4.3	8.3%	0.0	−2.2, 2.3	45.4%	−1.0	−4.2, 2.2
Cardiopathy									
Yes	49.0%	4.0 *	0.9, 7.2	8.8%	0.7	−1.6, 2.9	42.2%	−4.7 *	−7.7, −1.7
Diabetes									
Yes	52.3%	8.3 *	5.4, 11.2	8.3%	0.0	−2.0, 2.0	39.4%	−8.3 *	−11.0, −5.5
Worry about seasonal influenza									
Very/quite worried	53.9%	Ref.		14.5%	Ref.		31.6%	Ref.	
A little worried	44.1%	−9.8 *	−12.4, −7.2	7.0%	−7.5 *	−9.7, −5.4	48.9%	17.4 *	14.8, 19.9
Not worried	39.1%	−14.8 *	−18.1, −11.6	4.2%	−10.4 *	−12.7, −8.1	56.8%	25.2 *	22.0, 28.5
Perception of vaccine safety									
Very safe	53.4%	Ref.		7.9%	Ref.		38.7%	Ref.	
Quite safe	45.0%	−8.4 *	−11.3, −5.5	9.1%	1.2	−0.7, 3.2	45.9%	7.2 *	4.4, 9.9
Quite/very unsafe	34.5%	−18.9 *	−23.7, −14.2	7.2%	−0.7	−3.6, 2.3	58.4%	19.6 *	15.1, 24.1
Dear ones’ views on vaccination in general									
Very favorable	48.9%	Ref.		8.5%	Ref.		42.6%	Ref.	
Favorable	44.1%	−4.8 *	−7.7, −1.9	8.2%	−0.3	−2.4, 1.8	47.8%	5.1 *	2.3, 7.9
Quite favorable	44.2%	−4.7 *	−7.7, −1.6	8.5%	0.0	−2.2, 2.1	47.3%	4.7 *	1.8, 7.6
Quite unfavorable	44.1%	−4.8 *	−9.3, −0.3	7.9%	−0.6	−3.8, 2.5	48.0%	5.4 *	1.3, 9.5
Unfavorable/very unfavorable	48.1%	−0.7	−6.2, 4.7	7.2%	−1.3	−5.2, 2.6	44.6%	2.0	−2.7, 6.7
Awareness of having priority for vaccination									
Yes	52.1%	Ref.		8.1%	Ref.		39.8%	Ref.	
No	26.0%	−26.1 *	−29.5, −22.7	9.0%	0.9	−1.3, 3.1	65.0%	25.2 *	21.9, 28.5
Do not know	33.7%	−18.4 *	−22.2, −14.6	11.7%	3.6 *	0.8, 6.5	54.5%	14.7 *	11.4, 18.1
Advice to friends/relatives invited for vaccination									
Receive it	52.1%	Ref.		9.6%	Ref.		38.3%	Ref.	
Do not receive it	33.2%	−18.8 *	−23.9, −13.8	7.0%	−2.6	−5.4, 0.2	59.8%	21.4 *	16.5, 26.4
Do not know	24.3%	−27.8 *	−31.4, −24.2	6.1%	−3.5 *	−5.8, −1.3	69.6%	31.3 *	27.8, 34.8
Who invited for vaccination									
No one	30.0%	Ref.		9.7%	Ref.		60.4%	Ref.	
Local Healthcare Authority	48.0%	18.1 *	13.3, 22.9	5.9%	−3.7 *	−6.7, −0.8	46.0%	−14.3 *	−18.9, −9.8
Family doctor	50.2%	20.2 *	17.2, 23.2	8.0%	−1.7	−3.9, 0.6	41.8%	−18.6 *	−21.4, −15.7
Other medical doctors	48.3%	18.3 *	14.4, 22.3	10.5%	0.8	−2.1, 3.7	41.2%	−19.1 *	−22.8, −15.5

* *p*-value ≤ 0.05, that is, Δ significantly ≠ 0. ^†^ Including non-binary persons. *Abbreviations*: NUTS, Nomenclature of Territorial Units for Statistics.

## Data Availability

All data are provided within the manuscript.

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
