# Peer review of "Influenza Vaccine Uptake in Italy—The 2022–2023 Seasonal Influenza Vaccination Campaign in Italy: An Update from the OBVIOUS Project"

_vaccines, 2024, doi:10.3390/vaccines12030297_

Round 1

Reviewer 1 Report

Comments and Suggestions for Authors

This paper describes an update to a flu vaccine uptake study.  The data are clear and clearly presented.  And, although the results add nothing strikingly new to our understanding of vaccination uptake, they are of value to the community of scholars and policymakers.  I have very minor suggestions outlined below; my one more significant thought to enrich the paper is that the authors give their results some theoretical context; how do their results fit into models of human behavior and decision-making, or not?  With which approach are they most consistent?  For example, there’s a rich literature on social network theory applied to vaccination decisions.  Such contextualization is not just an intellectual exercise, but assists policymakers, brings the work to a wider audience, and helps readers have a better sense of how the data are meaningful or different.

Minor suggestions:

—it is not obvious what OBVIOUS stands for; please spell out this acronym

—line 58, ‘determines’ is problematic here; perhaps ‘demonstrates’ or ‘suggests’

—legend for Fig 2 need not only refer to Table 1’s elaboration of geographical areas of Italy, rather than repeating it all again

—line 235, ‘unaware of being pregnant’ is confusing, should be ‘unaware of pregnancy status’.

Reviewer 2 Report

Comments and Suggestions for Authors

The manuscript "INFLUENZA VACCINE UPTAKE IN ITALY The 2022–2023 seasonal influenza vaccination campaign in Italy: An update from the OBVIOUS project" deals with a very important topic, particularly after the covid-19 pandemic, when immunization became a topic of debate throughout society.

Vaccination, not only for influenza, but to prevent other diseases, has been reducing coverage in many countries, so this type of research is very timely.

Recommendations

[1] Introduction

The text of the introduction is adequate and makes the objectives of the research that resulted in this manuscript very clear. However, I recommend that the authors go deeper into the introduction by showing results of research on the flu vaccine in other European Union countries, for example. This will also improve the references in this manuscript, which I believe are few - which is why it needs more studies.

[2] Discussions

The authors present the results with excellence, however the discussions were poor. Based on the relevance of this study, it is essential that the authors in the discussion section carry out analyzes with recommendations that can guide public policy makers. Therefore, I recommend that these analyzes use comparisons with other studies related to the topic, which managed to expand vaccination coverage to prevent flu or other diseases. In Brazil, for example, the flu vaccination rate in many states has always achieved the desired coverage. I believe it is important for authors to deepen the discussion, this will allow the contributions of the manuscripts to be even more effective.

[3] Reference

The authors need to improve the references in the manuscript, as there are many studies in the world on this same topic.

Reviewer 3 Report

Comments and Suggestions for Authors

This is a large report based in a online form process in order to detect the causes of the low coverage for influenza in Italy. The main question is the approach to the analysis which cause a pulverized information in long tables with several sorting systems and a final aspects of using -percent for indicating their results. Tables are obscured by the large numbers of rows and columns with extensive statistical data and only asterisk for informing if the data were significant(very few). Figures are interesting specially figure 4 with important information on pregnant women, a very important risk groups for influenza, which is completely obscured by their data and statistical analysis was absent . It is obvious that the authors try to show all their data, but I believe that those descriptive studies are best presented in supplementary data and important data with statistics must be present in figure and table in the body text. If one is interested on the description the process of significance search, it could be found in supplementary data, but the reader must  be aware of important data, that cannot be obscured. 

In this way, each group at risk must be analysed briefly and not obscured or immersed by comparing with other groups. 

For example, the behavior of pregnant women must be discussed isolated for uptake of influenza vaccination.  There are any regional problems in this group ? Why they refuse vaccination at so large proportion? Focused in  groups at risk of severe influenza must improve the manuscript 
